# Impact of a Severe Dust Event on Diurnal Behavior of Surface Water Temperature in Subtropical Lake Kinneret

Pavel Kishcha [1,*] , Yury Lechinsky [2] and Boris Starobinets [1]

1   Department of Geophysics, Tel Aviv University, Tel Aviv 69978, Israel; starob@tauex.tau.ac.il
2   Kinneret Limnological Laboratory, Israel Oceanographic and Limnological Research, Migdal 1495000, Israel; yuryl@ocean.org.il
*   Correspondence: pavelk@tauex.tau.ac.il

**Abstract:** Dust impact on lake surface water temperature (SWT) over lakes, located in the Eastern Mediterranean, has not yet been discussed in previous publications. We investigated the effect of an extreme dust intrusion on the diurnal behavior of SWT in Lake Kinneret, appearing from 7–9 September 2015. This was carried out using METEOSAT and in-situ observations of SWT. In the presence of dust, METEOSAT SWT decreased along with increasing dust pollution both in the daytime and nighttime. This contradicted in-situ measurements of SWT at a depth of 20 cm which increased to 1.2 °C in the daytime and to 1 °C in the nighttime, compared to SWT on clear-sky September 6. The in-situ radiometer measurements of upwelling longwave radiation (ULWR) provided us with a criterion for assessing the reliability of METEOSAT and in-situ observations of SWT. Using this criterion, we found that, in the presence of dust, in-situ SWT was in line, whereas METEOSAT SWT contradicted in-situ ULWR. Considering in-situ ULWR is determined by actual SWT, we concluded that, in the presence of dust, in-situ SWT were capable of reproducing Kinneret SWT, while METEOSAT was incapable of doing so. An observed increase in daytime air temperature during the dust intrusion contributed to an increase in daytime Kinneret SWT. In the presence of maximal dust pollution on September 8, atmospheric humidity ($\rho_v$) exceeded by 30% that on clear-sky September 6. This increase in $\rho_v$ was observed in the absence of moisture advection indicating that dust intrusion can cause additional evaporation from Lake Kinneret and, consequently, intensify its drying up.

**Keywords:** fresh-water lakes; Lake Kinneret; surface water temperature; desert dust; dust intrusion; absolute atmospheric humidity; METEOSAT; upwelling longwave radiation

## 1. Introduction

Over the past half-century and especially during recent decades, the Eastern Mediterranean has been identified as one of the most prominent climate change hotspots, in accordance with Zittis et al. [1]. Several active deserts are located in this area, such as the Egyptian and Nubian deserts in the north-eastern Sahara, the Syrian-Iraqi desert, the Negev desert in Israel, and other deserts in the Arabian Peninsula. Observations and model studies have shown a general increase in desert dust pollution over the Eastern Mediterranean [1–5]. The area of the Sahara Desert is steadily increasing: it expanded by 10% over the twentieth century and is predicted to expand by a further 7% by the year 2050 [6,7]. This expansion of the Sahara is accompanied by increasing atmospheric dust pollution, as well as by the shrinking of productive lands in the Mediterranean coast of North Africa and of the Middle East.

Dust storms originating in the Eastern Mediterranean strongly affect the regional atmospheric radiation budget and air quality [8–10]. The radiative effect of dust results in a decrease in surface solar radiation (SR) causing the reduction of surface water temperature in such sea areas as the Red Sea [11] and the Persian Gulf [12]. With respect to lakes, located in the Eastern Mediterranean region, desert dust intrusions could influence their SWT by

the dust radiative effect. In addition, atmospheric dust particles absorb solar radiation causing them to become warmer: this leads to increasing longwave thermal radiation towards the lake surface [13]. Desert dust intrusions are accompanied by deposition of dust particles, in accordance with Kishcha et al. [14].

The subtropical fresh-water Lake Kinneret is in the northern part of the Jordan Rift valley, in Israel, in the Eastern Mediterranean region. Based on climate model predictions, La Fuente et al. [15] showed that this lake could disappear by the end of the 21st century as a result of decreasing precipitation and increasing evaporation. Kinneret SWT is one of the main factors determining evaporation. Dust intrusions can influence SWT. However, the dust impact on Kinneret SWT has not yet been discussed in previous publications. Therefore, a comprehensive investigation of the influence of dust intrusion on SWT in Lake Kinneret has become vital.

In this study, we investigated the effect of an extreme dust intrusion on the diurnal behavior of SWT in Lake Kinneret. This dust intrusion occurred in September 2015. Our study was carried out using 5 km × 5 km resolution hourly SWT records, from the geostationary METEOSAT satellites, supplemented by in-situ measurements of SWT.

## 2. Materials and Methods

### 2.1. Study Area

The fresh-water Lake Kinneret (Sea of Galilee) is in the northern section of the Jordan Rift valley, in Israel, in the Eastern Mediterranean, at ~210 m below sea level (b.s.l.): its maximal depth is ~40 m and surface area is ~166 km$^2$ (Figure 1) The lake is mainly fed by the north Jordan River, by runoff from the Golan Heights, and to a minor extent by underground springs [16]. Ziv et al. [17] found that, in the rainy season, Cyprus lows are the main cause of rainfall of ~400 mm per year over Lake Kinneret. Moreover, Cyprus lows are instrumental in the cooling of surface and epilimnion water in the lake, in accordance with Kishcha et al. [18].

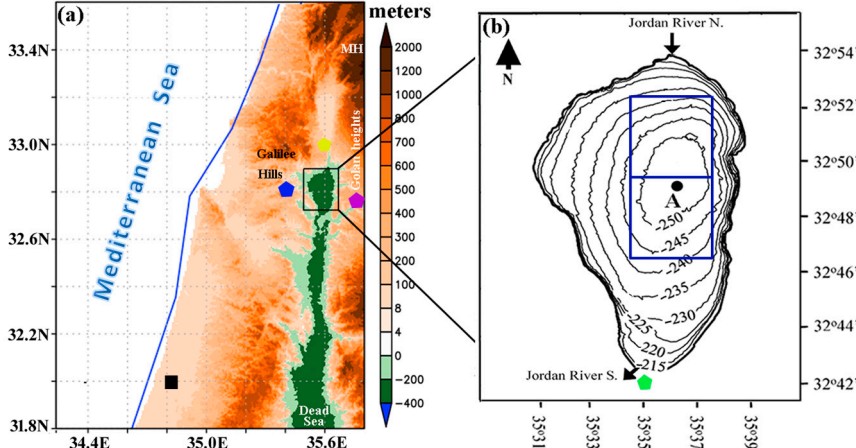

**Figure 1.** (**a**) Topographic map of the south-east Mediterranean region (30°N–33°N; 34°E–36°E) including Lake Kinneret, (**b**) a bathymetric map of Lake Kinneret (−215 to −250 m a.s.l.), The green, blue, yellow, and purple pentagons designate the location of the following stations: Zemah (32.70°N, 35.58°E), Deir Hanna (32.86°N, 35.37°E), Ayyellet Hashahar (33.02°N, 35.57°E), and Avne Etan (32.81°N, 35.76°E) respectively. The black square shows the location of the Bet Dagan (32.00°N, 34.81°E) station. MH designates Mount Hermon, while A (32.82°N, 35.60°E, 40 m depth) designates the location of the monitoring station conducting measurements of water temperature and meteorological parameters in the lake. The blue rectangles designate two pixels on the 0.05° × 0.05° METEOSAT grid which were not contaminated by land. These two pixels represent the water area in the lake (32.775°N–32.875°N; 35.575°E–35.625°E) where METEOSAT SWT was analyzed.

### 2.2. September 2015 Severe Dust Event in the Eastern Mediterranean Study Area

On 7 September 2015, one of the severest dust storms on record struck the Eastern Mediterranean and Israel in particular, as reported by Uzan et al. [19,20], Gasch et al. [21]. As illustrated in Figure 2, MODIS-Terra satellite imagery had started showing the presence of some amounts of dust pollution over the Jordan Rift valley already on September 7 at 10:30 LT, and significant amounts of dust pollution on September 8 and 9. This dust event was characterized by PM10 concentration of up to 3500 μg m$^{-3}$ on September 8 at 14 LT at the Afula monitoring site (39.59°N; 35.27°E) (compared to PM10 concentration less than 100 μg m$^{-3}$ on clear-sky September 6). This site is located at a distance of 40 km southwest from Lake Kinneret. (PM10 stands for particulate matter with an aerodynamic diameter of less than 10 μm.) According to Uzan et al. [20], on September 8, noon surface solar radiation dramatically dropped from ~900 W/m$^2$ to ~200 W/m$^2$ in the northern part of Israel. They found that the dust storm under study originated from desert regions in northern Syria: dust plumes penetrated Israel from northeast to southwest. Using ICON-ART model data, Gasch et al. [21] found that dust transport from Syria was responsible for the severe dust intrusion over Israel from 7–9 September 2015. Using ICON-ART model data, they showed that, on dusty days 7–8 September 2015, dust aerosol optical depth (AOD) ranged from 0.2 and 1.5 over the Zemah station near Lake Kinneret [21] (Figure 3).

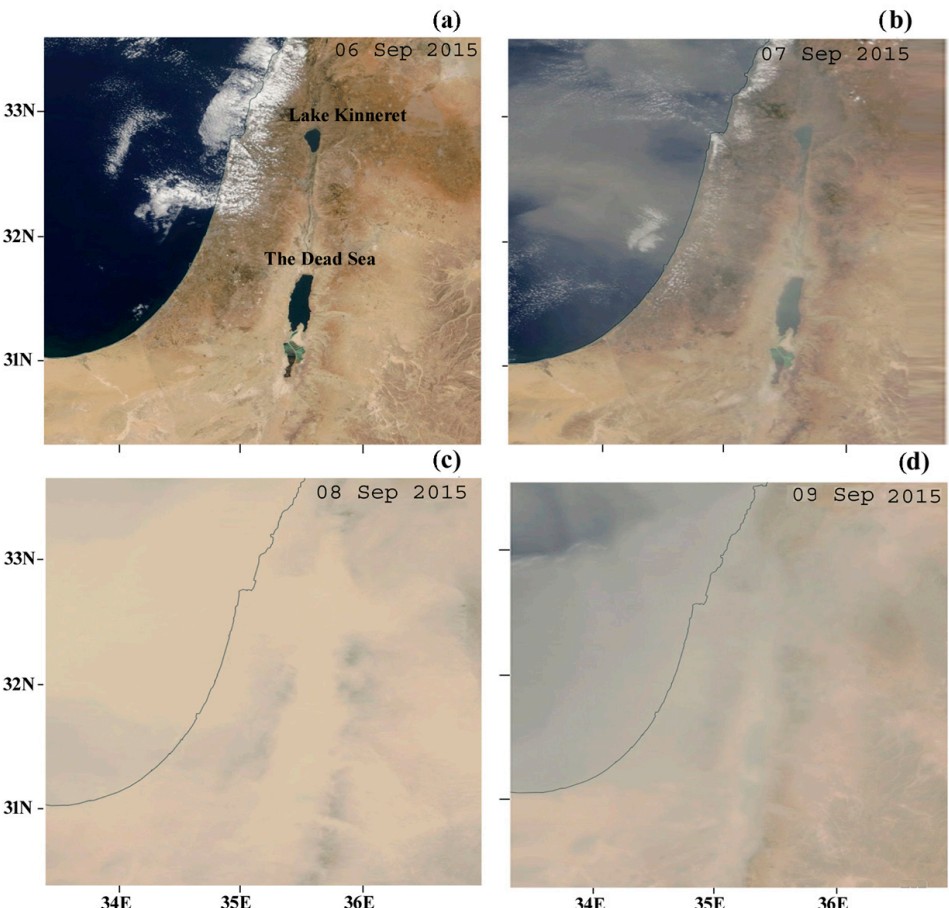

**Figure 2.** MODIS-Terra satellite imagery of the Jordan Rift valley and surrounding areas on (**a**) 6 September 2015 (**b**) 7 September 2015, (**c**) 8 September 2015, and (**d**) 9 September 2015, at 10:30 a.m. local time (LT).

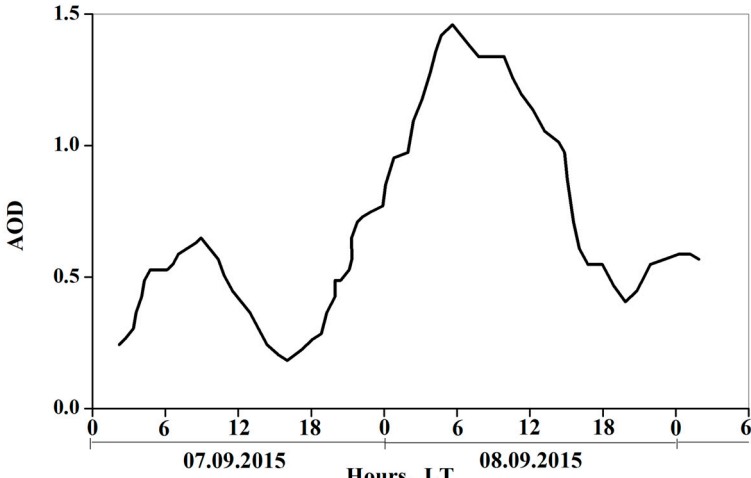

**Figure 3.** Diurnal variations of dust aerosol optical depth (AOD) on 7–8 September 2015, over the Zemah station (32.70°N, 35.58°E) near Lake Kinneret. Updated from [21].

### 2.3. Method

Our method to investigate the dust effect on diurnal variations of SWT was based on comparison: on comparing SWT on the dusty days 7–9 September 2015, with SWT under clear-sky conditions on September 6. In-situ radiometer measurements of upwelling longwave radiation (which is determined by actual SWT) provided us with a criterion for assessing the reliability of satellite (METEOSAT) and in-situ observations of SWT.

Furthermore, the dust intrusion caused changes in absolute atmospheric humidity ($\rho_v$) over the lake. To investigate the dust effect on diurnal variations of $\rho_v$ we used a similar method which was based on comparing $\rho_v$ on the dusty days 7–9 September 2015, with $\rho_v$ under clear-sky conditions on September 6.

### 2.4. Data

Similarly to our previous study by Kishcha et al. [22], diurnal variations of SWT in Lake Kinneret were analyzed using hourly data of METEOSAT land surface temperature (LST) records. These records were derived from the Spinning Enhanced Visible and Infra-Red Imager (SEVIRI) onboard the geostationary METEOSAT Second Generation satellites [23,24]. This product is presented on a 0.05° × 0.05° grid. The hourly METEOSAT LST data contain LST retrievals and their estimated uncertainty, according to the Product User Manual [25]. These model-based uncertainties can be used as quality indicators [23].

To avoid the effect of land contamination on METEOSAT LST retrievals, we focused on the Kinneret water area (32.775°N–32.875°N; 35.575°E–35.625°E): this area consists of two pixels on a 0.05° × 0.05° grid which were not contaminated by land (Figure 1). To estimate Kinneret SWT, we used METEOSAT SWT data averaging over these two pixels. Note that satellites measure SWT in the skin layer of 10–20 μm [26].

To complement the remotely sensed measurements of SWT, we used in-situ 10-min measurements of water temperature at Station A (St. A) at a depth of 20 cm (Figure 1) [27]. We consider that water temperature at a depth of 20 cm represents lake surface water temperature ($SWT_A$): this is due to vertical mixing in the lake. The $SWT_A$ measurements were taken using the Campbell 107-L temperature probe: its specifications are available online at https://www.campbellsci.asia/107-l (accessed on 7 November 2023). According to the specifications, the tolerance of in-situ measurements of $SWT_A$ was ± 0.2 °C for the temperature interval from 0 °C to 50 °C. Note that St. A is located near the boundary between the two METEOSAT pixels (Figure 1). Therefore, in this study, we compared in-situ measurements of $SWT_A$ with METEOSAT SWT averaged over the two pixels.

At the same location (St. A), 10-min measurements of wind speed (WS), air temperature (Tair) and relative humidity (RH) were performed together with measurements of

upwelling longwave (LW) radiation (4.5 to 42 μm) by the CNR4 Net Radiometer (specifications are available online at https://www.kippzonen.com/Product/85/CNR4-Net-Radiometer#.Y3XHy3ZByUk, accessed on 7 November 2023). The Campbell 107-L temperature probe and CNR4 Net Radiometer were calibrated by the manufacturer. The above meteorological measurements were taken at 2–3 m above the lake surface.

We converted relative humidity to absolute atmospheric humidity ($\rho_v$) using the approach by David et al. [28]:

$$\rho_v = 1324.45 \times \frac{RH}{100\%} \times \frac{\exp\left(\frac{17.67 \times Tair}{Tair + 243.5}\right)}{Tair + 273.15} \tag{1}$$

where RH stands for relative humidity (%), and Tair stands for air temperature (°C).

In addition, to analyze the presence of moisture advection, diurnal variations of absolute humidity were estimated at three other stations, located ~12 km from the lake in different directions, over the surrounding land areas: Ayyellet Hashahar (33.02°N, 35.57°E); Deir Hanna (32.86°N, 35.37°E); and Avne Etan station (32.81°N, 35.76°E) (Figure 1). This was carried out using 10-min meteorological measurements of 2-m air temperature and relative humidity taken at those stations.

To study dust radiative effects on surface solar radiation (SR), we used 10-min pyranometer measurements of global SR, direct SR, and diffuse SR. Measurements of global SR were taken at the following two meteorological stations: (a) Zemah (32.70°N, 35.58°E), located in the vicinity of Lake Kinneret; and (b) Bet Dagan (32.00°N, 34.81°E) in Central Israel (Figure 1). Measurements of direct and diffuse SR were taken at Bet Dagan. Solar radiation (0.285–2.880 μm) was measured by a Kipp & Zonen pyranometer CMP-11. The specifications of CMP-11 are available online at https://s.campbellsci.com/documents/eu/product-brochures/b_cmp11-l.pdf (accessed on 7 November 2023). For consistency with hourly METEOSAT SWT, the time series of all 10-min meteorological data as well as those of in-situ $SWT_A$ were processed as hourly means: to this end they have been smoothed by using one-hour running mean.

## 3. Results

### 3.1. Dust Influence on Solar Radiation

On September 7, the noon solar radiation (SR) was 800 W m$^{-2}$ compared to 870 W m$^{-2}$ on September 6, indicating the dust arrival (Figure 3). On September 8, when large amounts of dust arrived, the noon SR sharply decreased to 200 W m$^{-2}$. On September 9, the noon SR increased up to 630 W m$^{-2}$: this indicated a decrease in dust pollution (Figure 4).

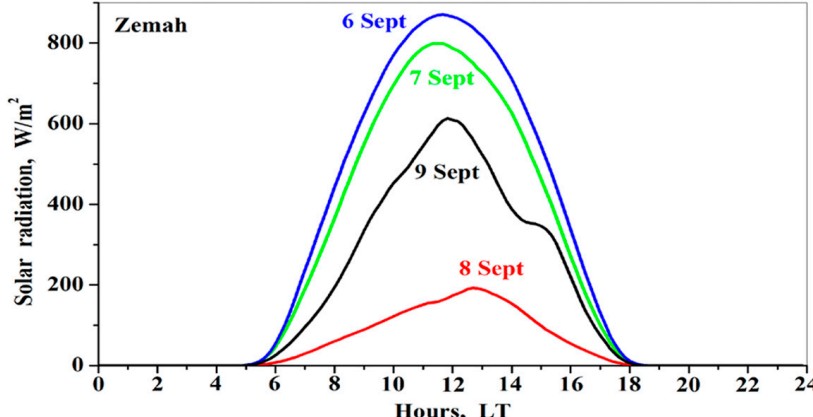

**Figure 4.** Diurnal variations of surface SR measured at the Zemah meteorological station on 6–9 September 2015.

The pyranometer at the Zemah station measures global SR, which was the sum of diffuse SR and direct SR. Separate measurements of direct and diffuse SR are conducted at

the Bet-Dagan station in Central Israel (Figure 1). Using those measurements in Bet-Dagan, we obtained the ratio ($\alpha$) of the noon value of direct SR to the noon value of diffuse SR, for every single day from September 6 to 9. Note that the dust event under study was a large-scale phenomenon: dust was more or less evenly distributed over Israel. This is supported by the similar day-to-day variations of SR observed separately at the Zemah and Bet-Dagan sites during the study period (September 6–9) (Figure 5). Therefore, to a first approximation, this ratio ($\alpha$) was the same as at the Bet-Dagan site and at sites near Lake Kinneret. Consequently, we used the obtained $\alpha$ values to estimate changes in noon direct and diffuse SR at the Zemah site near Lake Kinneret for the same dates from September 6 to 9 (Table 1).

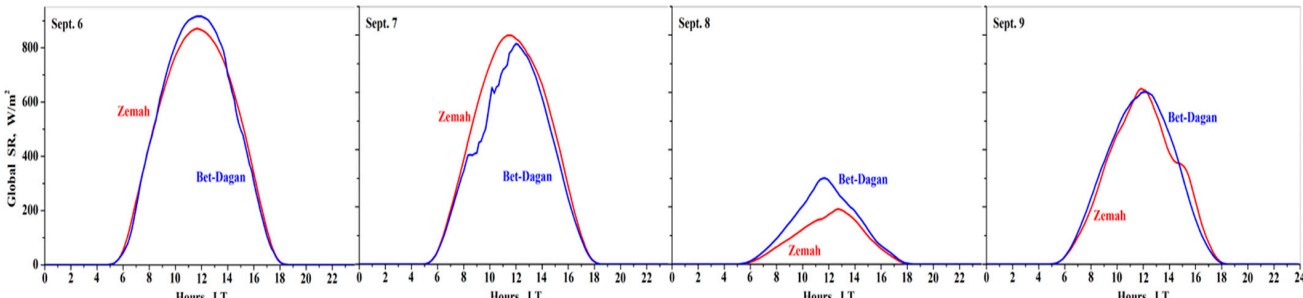

**Figure 5.** Comparison of day-to-day variations of global SR observed at (red lines) Zemah and (blue lines) Bet-Dagan stations located 100 km apart, during the study period (6–9 September 2015).

**Table 1.** The noon values of global solar radiation (GSR), diffuse solar radiation (DifSR), and direct solar radiation (DirSR) at the Zemah station on each day during the study period (6–9 September 2015). $\alpha$ designates the ratio of noon DirSR to noon DifSR.

| Date | GSR, W m$^{-2}$ | DifSR, W m$^{-2}$ | DirSR, W m$^{-2}$ | $\alpha$ |
|---|---|---|---|---|
| 6 September 2015 | 871 | 108 | 763 | 7.04 |
| 7 September 2015 | 802 | 321 | 481 | 1.50 |
| 8 September 2015 | 198 | 194 | 4 | 0.02 |
| 9 September 2015 | 636 | 509 | 127 | 0.25 |

We found that, compared to clear-sky September 6, dust on September 7 caused a decrease of ~40% in noon direct SR: this decrease (from 760 to 480 W m$^{-2}$) was observed along with a pronounced increase in noon diffuse SR (from ~100 to ~300 W m$^{-2}$) (Table 1). On September 8 in the presence of large amounts of dust, the noon direct SR was close to zero, whereas the observed noon global SR of ~200 W m$^{-2}$ consisted mainly of diffuse SR (Table 1). On September 9, the noon direct SR of 127 W m$^{-2}$ started recovering, although it was still four times lower than the noon diffuse SR (Table 1).

In accordance with the aforementioned data on noon global SR (GSR), direct SR (DirSR), and diffuse SR (DifSR), one can categorize amounts of dust on the dusty days under study as follows: maximal dust pollution took place on September 8 (GSR = 200; DirSR = 0; and DifSR = 200 W m$^{-2}$); intermediate dust pollution on September 9 (GSR = 640; DirSR = 130; and DifSR = 510 W m$^{-2}$); and low dust pollution on September 7 (GSR = 800; DirSR = 480; and DifSR = 320 W m$^{-2}$). Therefore, GSR decreased by approximately 10%, 30%, and 80% in the presence of small (September 7), intermediate (September 9), and maximal dust pollution (September 8) respectively: this was in comparison to GSR on clear-sky September 6.

## 3.2. Dust Influence on Upwelling LW Radiation

The in-situ radiometer measurements of upwelling longwave radiation (ULWR) (which is determined by actual SWT [29]) provided us with a criterion for assessing the

reliability of satellite (METEOSAT) and of in-situ observations of SWT. We analyzed variations in ULWR in order to find out information on characteristic features of SWT. This was carried out in four steps.

First, we compared ULWR on dusty September 7 and 9 with ULWR on clear-sky September 6. On dusty September 7 and 9, the daytime peak ULWR was higher than the daytime peak ULWR on clear-sky September 6 (Figure 6). This is an indication that, on September 7 and 9, daytime SWT was higher than daytime SWT on clear-sky September 6. This was despite the fact that, on those dusty days (September 7 and 9), global solar radiation (GSR) noticeably decreased.

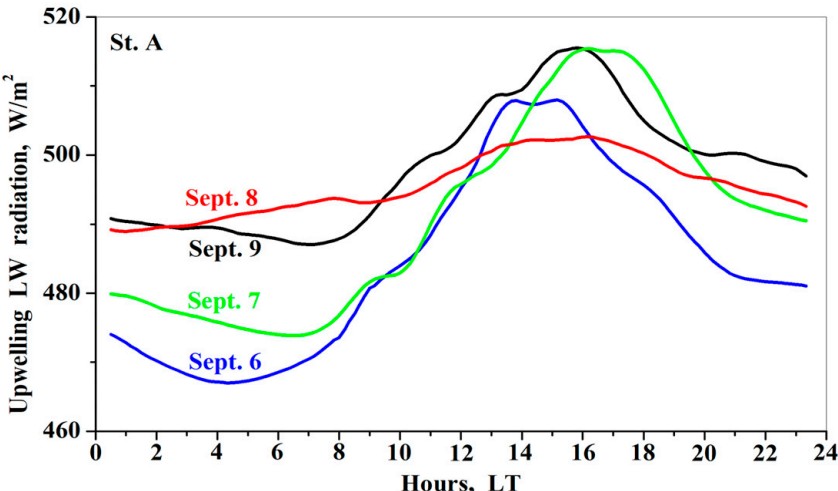

**Figure 6.** Diurnal variations of upwelling LW radiation on 6–9 September 2015.

Second, we compared ULWR on dusty September 8 with ULWR on clear-sky September 6. In the presence of maximal dust pollution on September 8, from 12 LT to 16 LT, the daily ULWR peak was lower than that on clear-sky September 6 (Figure 6). This is an indication that, due to the dust radiative effect on September 8, from 12 LT to 16 LT, SWT was lower than that on clear-sky September 6.

Third, we compared nighttime ULWR on all of the dusty days (September 7–9) with nighttime ULWR on clear-sky September 6. We found that nighttime ULWR on each dusty day was higher than ULWR on clear-sky September 6 (Figure 6). This indicates that dust pollution reflects part of ULWR back to the surface of the lake, leading to a noticeable increase in the nighttime SWT. As a result, nighttime SWT on all dusty days was higher than nighttime SWT on clear-sky September 6. It is worth adding that, on September 8 when dust amounts significantly increased (dust AOD increased from 0.5 to 1.5 (Figure 3)) during the period from 0 LT to 6 LT, ULWR increased in contrast to nighttime ULWR during the same period on the other days under study (Figure 6). This illustrates an increase in nighttime SWT along with an increase in dust intrusions. After sunset on that day (September 8), ULWR started decreasing in the nighttime, similarly to ULWR on the other days. This indicates a decrease in dust intrusions: AOD decreased to 0.5 (Figure 3).

And finally, we analyzed the daily range of ULWR (the difference between its daily maximum and minimum) on all days under study. We found that, on September 8, in the presence of maximal dust pollution, the ULWR daily range of ~30 W m$^{-2}$ was minimal compared to that on other dusty days. This is an indication that, on September 8, the daily range of SWT also was minimal.

*3.3. Dust Influence on Diurnal Variations of Satellite METEOSAT SWT*

In the absence of dust pollution on September 6, the diurnal variations of both METEOSAT SWT and in-situ ULWR were similar with respect to the following two points: (1) their daily peak at 14 LT, and (2) their decrease before sunrise (Figures 6 and 7). Therefore, in the absence of dust pollution, METEOSAT SWT was in line with in-situ ULWR.

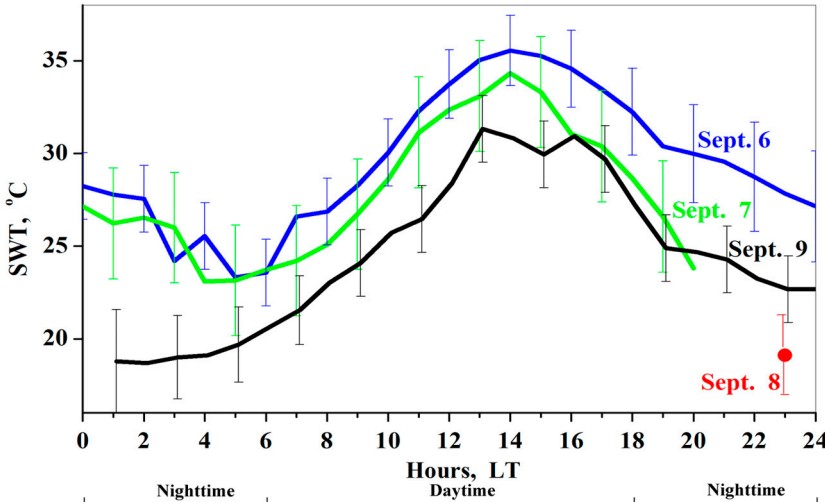

**Figure 7.** Diurnal variations of METEOSAT SWT on 6–9 September 2015. The vertical lines designate the uncertainty of SWT retrievals.

However, in the presence of dust, daily variations of METEOSAT SWT differed from daily variations of in-situ measured ULWR. On dusty September 7, there was a noticeable solar cycle in the diurnal variations of METEOSAT SWT with a peak at 14 LT, whereas ULWR peaked at 16–18 LT (Figures 6 and 7). On September 7, the METEOSAT peak was lower than that on clear-sky September 6, indicating a decrease in SWT in the presence of dust (Figure 7). In contrast, the ULWR peak of September 7 was higher than that on September 6. Considering that ULWR is determine by actual SWT, we concluded that METEOSAT showed erroneous daily variations of SWT even under comparatively low dust pollution on September 7.

On September 8, there were no METEOSAT observations of SWT due to the presence of large amounts of dust (dust AOD reached 1.5 at 5–6 LT (Figure 3)). The only measurement taken was at 23 LT, when SWT was equal to 19 °C (Figure 7): this SWT was lower by 9 °C than SWT on clear-sky September 6 at 23 LT (28 °C) (Figure 7).

The same low METEOSAT SWT of 19 °C was observed at hours from 0 LT to 2 LT on the following day, September 9. At 2 LT METEOSAT SWT started increasing and peaked at 13 LT (Figure 7). This contradicts in-situ ULWR measurements. On that day (September 9), in-situ ULWR decreased from 0 LT to 7 LT, then it started increasing and peaked at 16–18 LT (Figures 6 and 7).

It is worth mentioning that the METEOSAT SWT uncertainties on dusty days September 7–9 and on clear sky day September 6 are about the same (Figure 7). METEOSAT data showed that daytime SWT decreased along with increasing dust pollution: daytime SWT on September 7 (low dust pollution) was lower than that on clear-sky September 6 and higher than daytime SWT on September 9 (intermediate dust pollution) (Figure 7). In contrast, in-situ ULWR measurements showed that, on dusty days September 7 and 9, daytime ULWR were higher than that on clear-sky September 6 (Figure 6). Considering that in-situ ULWR is determined by actual SWT, the above information indicates that, in the presence of dust, METEOSAT SWT was incapable of reproducing diurnal behavior of Kinneret SWT.

*3.4. Dust Influence on Diurnal Variations of In-Situ Measured SWT$_A$*

We analyzed diurnal variations of SWT$_A$ measured in-situ at St. A, at a depth of 20 cm. We consider that water temperature at a depth of 20 cm represents lake surface water temperature. This is due to vertical mixing in the lake even under weak winds (Figure 8). We used the obtained daily variations of in-situ ULWR to examine if in-situ SWT$_A$ was capable of reproducing Kinneret SWT.

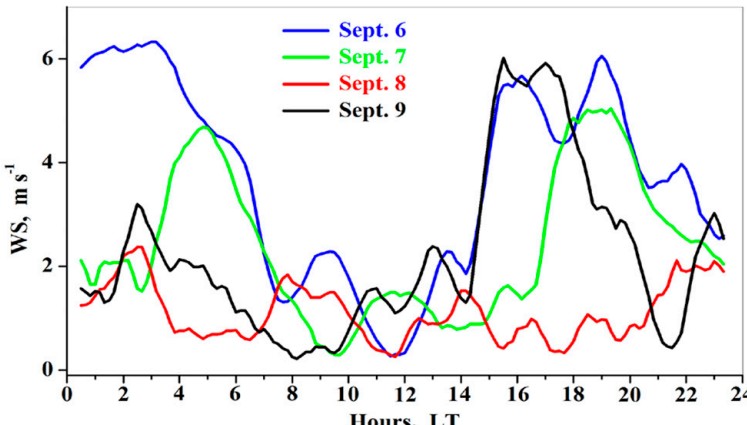

**Figure 8.** Diurnal variations of wind speed (WS) measured at St. A on 6–9 September 2015.

On clear-sky September 6, $SWT_A$ peaked at ~13:30 LT; it was in line with in-situ ULWR (Figures 6 and 9). On the following day (September 7, when GSR decreased by ~10% as a result of dust intrusions), one could expect a decrease in $SWT_A$. In-situ measurements showed that, after dust arrival, $SWT_A$ started increasing: the exceedance varied from 0.5 to 1.2 °C in the daytime (13:30 LT to 18 LT) in comparison with daytime $SWT_A$ on clear-sky September 6 (Figure 9). It is worth mentioning that the observed exceedance of daytime $SWT_A$ on dusty September 7 over daytime $SWT_A$ on clear-sky September 6 was in line with in situ radiometer measurements of ULWR (Figure 6). This fact highlights that in-situ $SWT_A$ (measured at a depth of 20 cm) could reproduce the characteristic features of daytime Kinneret SWT in the presence of dust pollution on September 7.

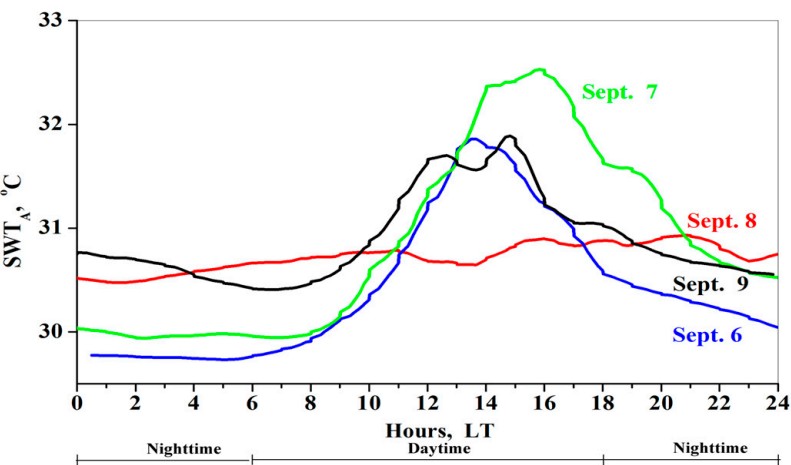

**Figure 9.** Diurnal variations of in-situ measured $SWT_A$ at St. A, at a depth of 20 cm, on 6–9 September 2015.

On September 7, an increase in daytime $SWT_A$ was accompanied by an increase in air temperature (Tair). Specifically, starting from 13 LT on September 7, Tair was mainly higher than that on September 6 during the same time period (Figure 10a). The increase in Tair contributed to the noticeable increase in $SWT_A$ after dust arrival on September 7. This fact (that the increase in Tair was accompanied by the increase in daytime $SWT_A$) is evidence that, on September 7, water heating by the near-ground atmospheric layer (which is in contact with the lake water surface) was more intense than water cooling due to the dust radiative effect, in the presence of low dust pollution.

On September 8, in the presence of maximal dust pollution, there was no noticeable solar cycle in the diurnal variations of $SWT_A$: $SWT_A$ was almost constant throughout the day (Figure 9). This was in line with ULWR measurements, which showed that, on September 8, the daily range of ULWR (~30 W m$^{-2}$) was minimal compared to the

daily ULWR range on the other days under study. This fact highlights that in-situ SWT$_A$ measurements could reproduce characteristic features of Kinneret SWT in the presence of maximal dust pollution on September 8.

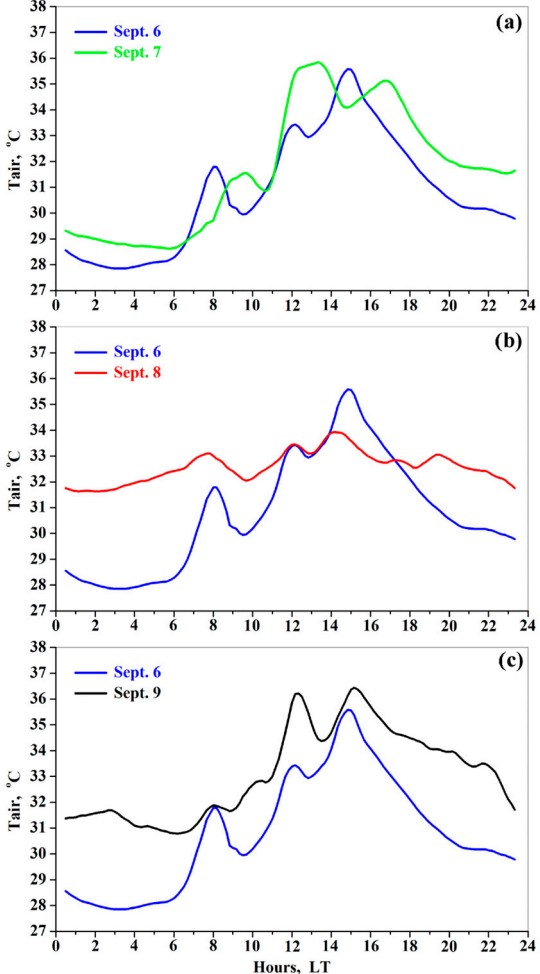

**Figure 10.** Diurnal variations of air temperature (Tair) measured at St. A in the lake on (**a**) 6 and 7 September 2015; (**b**) 6 and 8 September 2015; and (**c**) 6 and 9 September 2015.

On September 8, in the daytime, water cooling by the dust radiative effect was more intense than water heating by the near-ground atmospheric layer. This was confirmed by the decrease in SWT$_A$ from 11 LT–17 LT on that day, when SWT$_A$ was lower by ~1 °C than SWT$_A$ on clear-sky September 6 (Figure 9). In the nighttime on September 8, SWT$_A$ was higher by ~1 °C than SWT$_A$ on clear-sky September 6. This was in line with ULWR measurements, which showed that nighttime ULWR on each dusty day was higher than nighttime ULWR on clear-sky September 6.

On September 9, in the presence of intermediate dust pollution, there was a noticeable solar cycle in the diurnal variations of SWT$_A$ (Figure 9). In the nighttime, on that day (September 9), SWT$_A$ was higher by up to 1 °C than SWT$_A$ on clear-sky September 6, in line with ULWR measurements (Figure 6).

### 3.5. Dust Effect on Atmospheric Humidity

To estimate the impact of the dust intrusion on absolute atmospheric humidity ($\rho_v$) over Lake Kinneret, we used in-situ measurements of relative humidity (Figure 11) and of atmospheric temperature (Figure 10), taken at St. A during the study period (6–9 September 2015). We converted relative humidity to absolute humidity ($\rho_v$) using formula (1), which is based on the approach by David et al. [28]. Unexpectedly, measurements showed that

on September 8, in the absence of direct solar radiation, $\rho_v$ increased up to 28.7 g m$^{-3}$ at 17 LT, which was higher than that on clear-sky September 6 (24 g m$^{-3}$) (Figure 12). On September 9, before sunrise, the same phenomenon of increasing $\rho_v$ was observed despite weak winds (<2 m s$^{-1}$) (Figure 12). After sunset on that day (September 9), $\rho_v$ was lower than on September 6 (Figure 12). This decrease in $\rho_v$ on September 9 after sunset could be explained by the arrival of dry air, as illustrated by a decrease in relative humidity during that period (Figure 11).

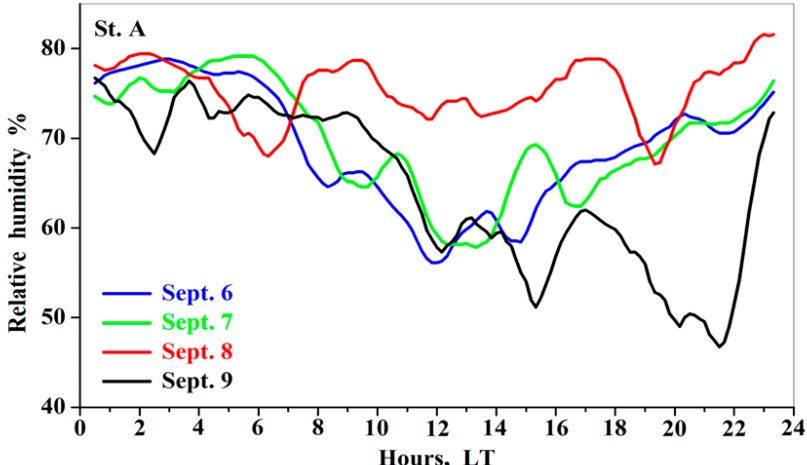

**Figure 11.** Diurnal variations of relative humidity measured at St. A in the lake on 6–9 September 2015.

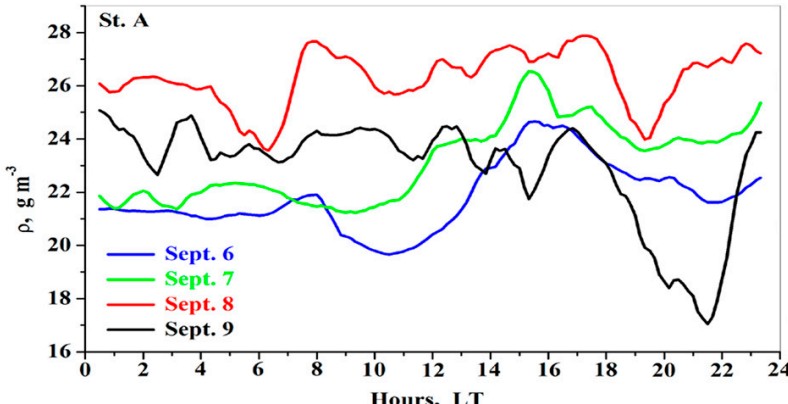

**Figure 12.** Diurnal variations of absolute humidity ($\rho_v$) measured at St. A in the lake on 6–9 September 2015.

To establish a relationship between an increase in $\rho_v$ at St. A in the lake and amounts of dust on each dusty day under study (Section 3.1), we conducted a ratio analysis. We analyzed diurnal variations of the ratio (in %) of $\rho_v$ on dusty days September 7–9 and that on clear-sky September 6. The analysis showed that, on September 8, in the presence of maximum dust pollution, the ratio reached 30% from 8 LT to 13 LT (Figure 13). On September 9, in the presence of intermediate dust pollution, the ratio reached 20%, and, on September 7, in the presence of low dust pollution, the ratio reached only 15% (Figure 13).

As illustrated in Figure 14, similar ratio analyses at three other meteorological stations (B, C, and D), located ~12 km from the lake in different directions, showed a totally different picture compared to St. A. Specifically, at each of the aforementioned stations, there was no increase in $\rho_v$ but even a decrease in $\rho_v$ was observed on September 8 (in the presence of maximal dust pollution) compared to that on clear-sky September 6. This is evidence that the observed increase in $\rho_v$ at station A in the lake was not caused by any advection of

moist air but was a result of increased evaporation from the lake surface in the presence of dust.

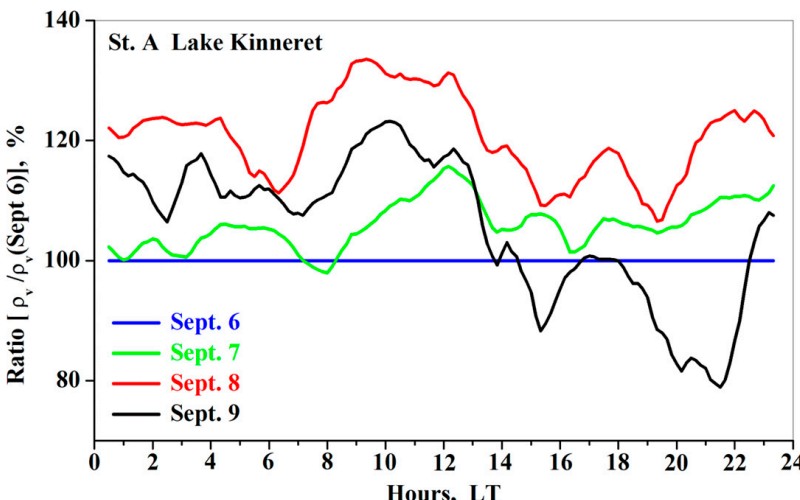

**Figure 13.** Diurnal variations of the ratio (in %) of atmospheric absolute humidity on dusty days 7–9 September 2015 and on clear-sky 6 September 2015, at St. A located in the lake (32.82°N, 35.60°E).

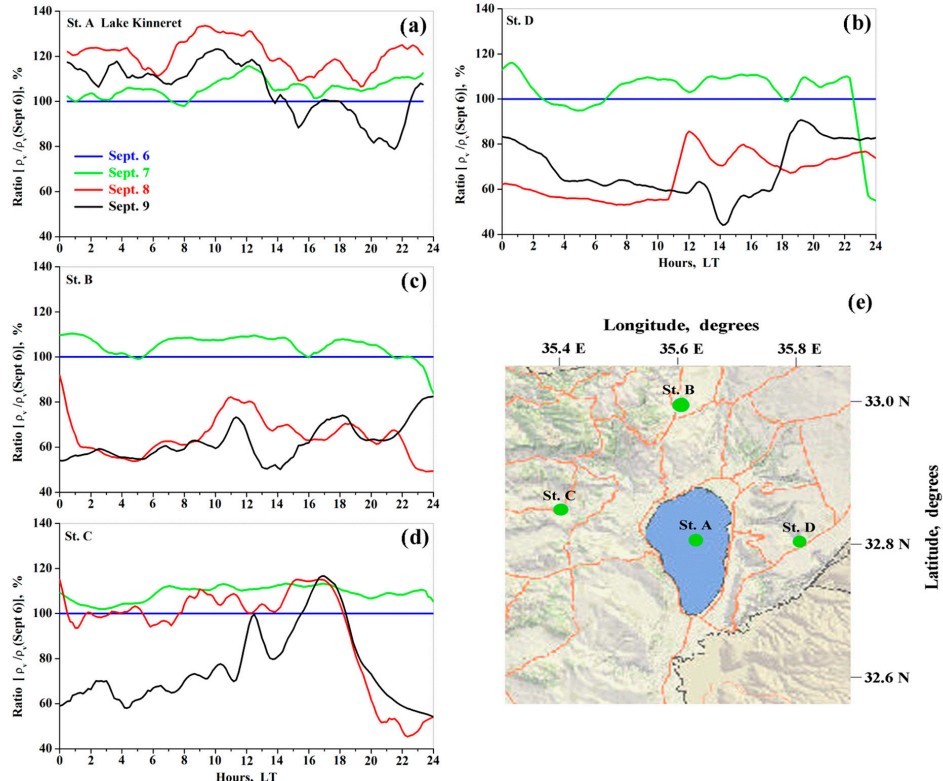

**Figure 14.** Diurnal variations of the ratio (in %) of atmospheric absolute humidity on dusty days 7–9 September 2015 and on clear-sky 6 September 2015, at four meteorological stations: (**a**) St. A located in the lake (32.82°N, 35.60°E); (**b**) St. B represents the Ayyellet Hashahar station (33.02°N, 35.57°E); (**c**) St. C—the Deir Hanna station (32.86°N, 35.37°E); and (**d**) St. D—the Avne Etan station (32.81°N, 35.76°E). (**e**) a map of the Kinneret region with the location of meteorological stations. Stations B, C, and D are ~12 km from the lake.

## 4. Discussion

A strong dust intrusion occurred over Lake Kinneret from 7–9 September 2015—when global solar radiation (GSR) decreased from 870 to 200 W m$^{-2}$, due to a dust radiative effect. This decrease in GSR should have led to a decrease in Kinneret SWT in the daytime. Indeed, METEOSAT showed that SWT decreased along with increasing dust pollution (Tables 2 and 3, Figure 7).

**Table 2.** Maximum (in absolute value) temperature difference between in-situ SWT$_A$, METEOSAT SWT, Tair on each dusty day (September 7–9) and SWT$_A$, SWT, Tair on clear-sky September 6, in the daytime (10 LT–16 LT) when solar radiation was the strongest.

| Date | Δ SWT$_A$ [°C] | Δ SWT [°C] | Δ Tair [°C] |
|---|---|---|---|
| 7 September 2015 | 1.2 | −3.5 | 3.2 |
| 8 September 2015 | −1.0 | (−9.0) * | −2.9 |
| 9 September 2015 | 0.5 | −5.8 | 4.4 |

* On September 8, in the present of maximal dust pollution, METEOSAT SWT was available only at 23 LT. One can suggest that approximately the same low SWT was observed throughout that day (September 8), in accordance with the almost constant in-situ SWT$_A$.

**Table 3.** Maximum (in absolute value) temperature difference between in-situ SWT$_A$, METEOSAT SWT, Tair on each dusty day (September 7–9) and SWT$_A$, SWT, Tair on clear-sky September 6, in the nighttime (0 LT–6 LT and 18 LT–24 LT).

| Date | Δ SWT$_A$ [°C] | Δ SWT [°C] | Δ Tair [°C] |
|---|---|---|---|
| 7 September 2015 | 1.1 | −6.2 | 2.2 |
| 8 September 2015 | 0.9 | −9.0 | 4.3 |
| 9 September 2015 | 1.0 | −9.0 | 4.1 |

The reliability of satellite-based SWT observations in the presence of dust pollution was discussed in previous publications. Al-Shehhi [30] found that satellite sea surface temperature estimates are influenced by the instantaneous changes in dust levels, particularly when the aerosol optical depth (AOD) exceeds 0.3. During the dust event under study in the current paper, dust AOD ranged from 0.2 to 1.5 over Lake Kinneret (Figure 3). Such a significant variability of dust AOD points to irregular dust intrusions. On September 8 when AOD exceeded 0.7, METEOSAT SWT retrievals were unavailable. However, on the other dusty days METEOSAT SWT retrievals together with their estimated uncertainties were available. As discussed below, in the presence of dust, the quality of available SWT retrievals requires further examination using ground-based measurements.

We found that, in the absence of dust pollution on September 6, both METEOSAT SWT and in-situ SWT at a depth of 20 cm were in line with each other. However, in the presence of dust, METEOSAT SWT decreased along with increasing dust pollution both in the daytime and nighttime. In contrast, in-situ SWT measurements showed even an increase in SWT in the daytime and nighttime. The in-situ radiometer measurements of upwelling longwave radiation (ULWR), which is determined by actual SWT, provided us with a criterion for assessing the reliability of METEOSAT and in-situ observations of SWT. Using this criterion, we found that, in the presence of dust, in-situ SWT was in line, whereas METEOSAT SWT contradicted in-situ ULWR. This finding led us to the conclusion that, in the presence of dust, in-situ SWT was capable of reproducing Kinneret SWT, while METEOSAT was incapable of doing so.

In-situ measurements of SWT and ULWR showed that the impact of the severe dust intrusion on Kinneret SWT in the daytime and in the nighttime was as follows:

In the daytime on dusty days September 7 and 9, increasing dust pollution caused an increase (up to 1.2 °C) in SWT and an increase in ULWR: this was in comparison to daytime SWT and ULWR on clear-sky September 6 (Figures 6 and 9). Such a phenomenon

(an increase in daytime SWT caused by increasing dust pollution) has not been discussed in previous publications. This phenomenon can be explained as follows. On those dusty days (September 7 and 9), in the daytime, increasing dust pollution caused an increase in $T_{air}$ by up to 3.2 °C and 4.4 °C correspondingly, compared to daytime $T_{air}$ on clear-sky September 6 (Table 2). This air heating in the near-ground atmospheric layer (which is in contact with the lake water surface) contributed to water heating in the lake. This water heating by the near-ground atmospheric layer was more intense than water cooling due to the dust radiative effect.

However, in the presence of maximal dust pollution on September 8 (when GSR decreased by ~80%), in the daytime, the dust radiative effect caused a decrease in $T_{air}$ by up to 2.9 °C, compared to daytime $T_{air}$ on clear-sky September 6 (Table 2). This air cooling in the near-ground atmospheric layer contributed to a decrease in daytime SWT by up to 1 °C, compared to daytime SWT on clear-sky September 6 (Table 2).

As for the nighttime on dusty days September 7–9, in-situ measurements showed that an increase in $T_{air}$ up to 4.3 °C was accompanied by an increase in SWT up to 1 °C, compared to nighttime $T_{air}$ and SWT on clear-sky September 6 (Table 3). This was in line with ULWR measurements, which showed that nighttime ULWR on each dusty day under study was higher than nighttime ULWR on clear-sky September 6 (Figure 6). This is evidence that dust pollution reflects part of ULWR back to the surface of the lake, leading to a noticeable increase in the nighttime SWT. It is worth adding that, on September 8 when dust amounts significantly increased (dust AOD increased from 0.5 to 1.5 (Figure 3)) during the period from 0 LT to 6 LT, ULWR increased in contrast to ULWR during the same period on the other days under study (Figure 6). This illustrates an increase in nighttime SWT along with increasing dust intrusions on September 8. Note that this increase in nighttime SWT was accompanied by an increase in nighttime $T_{air}$ contributing to surface water heating (Figure 10b).

In this study, we also analyzed the impact of dust pollution on absolute atmospheric humidity ($\rho_v$). In the presence of maximal dust pollution on dusty September 8, the maximal $\rho_v$ over the lake was observed: $\rho_v$ became higher by 30% compared to $\rho_v$ on clear-sky September 6. This maximal increase in $\rho_v$ on September 8 was observed in the absence of moisture advection.

We investigated the relationship between dust amounts and an increase in absolute humidity. Our findings showed that the increase in $\rho_v$ reached 30%, 20%, and 15% in the presence of maximum, intermediate, and low dust pollution on September 8, 9, and 7 respectively: this was in comparison to $\rho_v$ on clear-sky September 6. Considering gravitational settling of dust particles, this finding indicates a relationship between dust deposition on Kinneret water surface and an increase in absolute humidity ($\rho_v$). We consider that, during the dust intrusion, settling of a great amount of dust particles on the Kinneret water surface could have caused a decrease in surface water tension. Agrawal and Menon [31] investigated the relationship between water surface tension and evaporation. Their findings imply that a decrease in water surface tension leads to an increase in evaporation, and consequently, to an increase in absolute humidity.

## 5. Conclusions

Climate model predictions showed that Lake Kinneret could disappear by the end of the 21st century due to decreasing precipitation and increasing evaporation [15]. Kinneret SWT is one of the main factors determining evaporation. During the last several decades, observations and model data showed increasing desert dust pollution over the Eastern Mediterranean. The dust impact on Kinneret SWT has not yet been discussed in previous publications. In this study, we showed that dust intrusions can influence Kinneret SWT.

We investigated the effect of an extreme dust intrusion on the diurnal behavior of SWT in Lake Kinneret, which appeared from 7–9 September 2015. This was carried out using satellite METEOSAT and in-situ observations of SWT. In the presence of dust, METEOSAT showed that SWT decreased along with increasing dust pollution both in the daytime

and nighttime. This contradicted in-situ measurements of SWT at a depth of 20 cm which showed an increase up to 1.2 °C in the daytime (despite the dust radiative effect) and up to 1 °C in the nighttime: this was in comparison to daytime and nighttime SWT on clear-sky September 6. The in-situ radiometer measurements of upwelling longwave radiation (ULWR) provided us with a criterion for assessing the reliability of METEOSAT and in-situ observations of SWT. Using this criterion, we found that, in the presence of dust, in-situ SWT was in line, whereas METEOSAT SWT contradicted in-situ ULWR. Considering that in-situ ULWR is determined by actual SWT, we concluded that, in the presence of dust, in-situ SWT measurements were capable of reproducing Kinneret SWT, while METEOSAT was incapable of doing so.

An observed increase in daytime air temperature on dusty days September 7 and 9 contributed to an increase in daytime Kinneret SWT compared to daytime SWT on clear-sky September 6. However, a decrease in daytime air temperature on September 8 (in the presence of maximal dust pollution) contributed to a decrease in daytime Kinneret SWT.

As for the nighttime on dusty days September 7–9, in-situ measurements showed that an increase in Tair up to 4.3 °C was accompanied by an increase in SWT up to 1 °C, compared to nighttime Tair and SWT on clear-sky September 6. This was in line with ULWR measurements, which showed that nighttime ULWR on each dusty day under study was higher than nighttime ULWR on clear-sky September 6. This is evidence that dust pollution reflects part of ULWR back to the surface of the lake, leading to a noticeable increase in nighttime SWT. It is worth adding that, on September 8 when dust amounts significantly increased (dust AOD increased from 0.5 to 1.5) during the period from 0 LT to 6 LT, ULWR increased in contrast to ULWR during the same period on the other days under study. This illustrates an increase in nighttime SWT along with increasing dust intrusions on September 8. Note that this increase in nighttime SWT was accompanied by an increase in nighttime Tair contributing to surface water heating.

Note that, in this study, available METEOSAT SWT retrievals were used together with their estimated uncertainty (Figure 7) [23,25]. We found that, in the presence of dust, the quality of available METEOSAT SWT retrievals requires further examination using in-situ measurements.

During the dust intrusion, a noticeable increase in $\rho_v$ over the lake was observed: $\rho_v$ reached 30%, 20%, and 15% in the presence of maximum, intermediate, and low dust pollution on September 8, 9, and 7 respectively: this was in comparison to $\rho_v$ on clear-sky September 6. The maximal increase in $\rho_v$ on September 8 was observed in the absence of moisture advection: this indicates that dust intrusion can cause additional evaporation from Lake Kinneret. This finding implies the following significant point: increasing desert dust pollution over the Eastern Mediterranean can intensify the drying up of Lake Kinneret.

**Author Contributions:** All co-authors equally contributed to the writing of the current research article: P.K., Y.L. and B.S.; in-situ measurements of water temperature and ULWR in Lake Kinneret: Y.L. All authors have read and agreed to the published version of the manuscript.

**Funding:** This research received no external funding.

**Data Availability Statement:** Satellite METEOSAT product of land surface temperature (physical model) is available at https://wui.cmsaf.eu/safira/action/viewProduktHome (accessed on 7 November 2023). In-situ surface water temperature measurements, taken at a depth of 20 cm at site A, together with meteorological measurements are available in Zenodo repository at https://doi.org/10.5281/zenodo.7682144 (accessed on 7 November 2023). Pyranometer data of solar radiation taken at the meteorological stations Zemah and Bet-Dagan are available at https://ims.gov.il/en/data_gov (accessed on 7 November 2023). Measurements of 2-m air temperature and relative humidity at meteorological stations located in North Israel are available at https://ims.gov.il/en/data_gov (accessed on 7 November 2023).

**Acknowledgments:** We thank the Satellite Application Facility on Climate Monitoring (CM SAF) team for METEOSAT LST records. We thank the Israel Meteorological Service for 10-min meteorological and pyranometer measurements during our study period. In-situ measurements of water temperature and ULWR at site A in Lake Kinneret are associated with the Kinneret Limnological Laboratory, Israel oceanographic and Limnological Research (https://www.ocean.org.il, accessed on 13 August 2023).

**Conflicts of Interest:** The authors declare no conflict of interest.

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
