# Peer review of "Impact of a Severe Dust Event on Diurnal Behavior of Surface Water Temperature in Subtropical Lake Kinneret"

_remotesensing, doi:10.3390/rs15225297_

Round 1
Reviewer 1 Report (Previous Reviewer 1)
Comments and Suggestions for Authors
The manuscript is intersting and covering an important topic on the impact of severe dust storms on surface water temperature which also have a bearing on water quality and freshwater availability. I have the following comments which I believe will improve the quality of the manuscript further;
- in the abstract in line 17, remove ":the higher the dust pollution the lower SWT". This in sense is an unnecessary repetition or emphasis of what previous words in the same sentence are already saying. The point is already clear without that
- I found line 18 to 24 of the abstract as diverting from the focus of the manuscript. The focus must be on impact of dust storms and not on comparison of in-situ measurements with satellite retrieval. The two datasets must have been combined to improve expression of impacts rather than comparing them. If the researchers interest was on accuracy of measured impacts they should have used in-situ observations to improve remotely sensed measurements of impacts. In such a study, remote sensing must not be competing but complementing in-situ measurements
- Figure 2 supports the point raised above. Clearly, remote sensing can depict the impact of dust storms as patterns
Author Response
See the attached file.

Reviewer 2 Report (Previous Reviewer 2)
Comments and Suggestions for Authors
The Eastern Mediterranean is subject to an increase in desert dust pollution over the last several decades. Consequently, there are accordingly responses of surface water temperature and atmospheric humidity to this increased dust pollution condition. Using satellite and in-situ observations, this study investigated the influence of a severe dust event on the diurnal behavior of surface water temperature and absolute humidity in the Kinneret Lake. The paper is a re-submission, and has significantly improved. I would like to recommend it for publication after minor revision.
1. Abstract, Line 24-25, “Sept.” > “September” in abstract.
2. Introduction, Line 55, “Kishcha et al. [14]” > “[14]”.
3. Introduction, Line 59, 21 st > 21st, should not be upper case.
4. Section 2.1, Line 74, what means b.s.l.?
5. Section 2.2, Line 103, “LT” > “local time (LT)”
6. Section 2.2, Line 105, “14 LT” > “14:00 LT”
7. Section 2.2, Line 109, “um.)” > “um).”
8. Section 2.4, Line 156-157, “… for the temperature range of 0-50 °C”
9. Section 2.4, Line 158-159, when comparing the in-situ SWTA with METEOSAT SWT, is it necessary to adjust them at the same depths? Because in-situ SWT were measured at 20 cm water depth, but the METEOSAT SWT were the skin temperature.
10. Equation (1), “T” > “Tair”, maybe better to keep it consistent with the text Line 162.
11. Section 3, Figure 3-4, maybe better to mark the stages of the dust event in the figure, e.g., “clear-sky”, ‘dust arrival’, …
12, Section 3, Figure 6, why there are double peaks on Sept. 9 at 13:00 and 16:00, respectively?
13, Section 3, Line 329, “18 LT” > “18:00 LT”
14, Section 3, Line 330, “14 LT” > “14:00 LT”
15, Section 3, Line 335-337, why the in-situ SWTA was in contrast to METEOSAT SWT?
16, Some of the figures, the label of x-axis, are “Hours, LT”, but some are (LT, Hours), please revise. Day-time and night-time should also be marked for better understanding. For example, shading the night-time hours in the figures.
17, Wind speed should also important for the SWT, evaporation, maybe the role of wind speed can be further discussed in Figure 7.
Comments on the Quality of English LanguagePlease see Comments and Suggestions for Authors
Author Response
See the attached file.

Reviewer 3 Report (Previous Reviewer 3)
Comments and Suggestions for Authors - Overall:
This is the resubmission of the manuscript that analysed the patterns of the water surface temperature of a lake during the passage of a dust storm. Despite replying to some, but not all, of my previous comments, I do not think there is significant improvement in the manuscript. The results are not of significant interest for the theme. The thermal sensor is not correctly applied in this study, and if there is interesent in understanding the effects of the dust storm on lake evaporation, please use a robust methodology for that, even if for just a point (where there is water temperature data) on the lake. Below there are further comments:
- Major issues:
It was clear that it is not possible to use METEOSAT, this is not a reasonable result! A long-wave thermal sensor cannot penetrate obstacles, the satellite is misused and should not be included.
The patterns of evaporation during the passage of the dust storm should be actually tested with calculating the latent heat flux during this short period of time. There is enough data for that, and the results would be of much more relevance.
Again, the plots used in the manuscript are not suitable for publication in a respected scientific journal.
Author Response
See the attached file.

Reviewer 4 Report (Previous Reviewer 4)
Comments and Suggestions for Authors
The authors have addressed my concerns and I believe the manuscript is ready for publication.
Author Response
See the attached file.

This manuscript is a resubmission of an earlier submission. The following is a list of the peer review reports and author responses from that submission.
Round 1
Reviewer 1 Report
Comments and Suggestions for Authors
- the first sentence of the abstract must be on the motivation or background of the research
- the abstract goes straight to the findings without briefing on the methods used
- could also conclude abstract by statement on application of the findings
- the introduction is not indicating the gap being addressed by the study. What is the novelty of the study?
- I suggest linking the study area to continent boundary for context. In the currect state, the readers' knowledge is over-assumed
- in figure 5, a key is needed to show what the red and blue lines are representing
- in figure 9, there no line but just a red dot for 8 September
- besides comparing the days using lines, statistical analysis can improve the findings. For example ANOVA to show that the radiation amounts are statistically significantly different at different levels of dust
- the discussion sounds more of a repetition of results. Implications of the findings need to be thoroughly discussed
Reviewer 2 Report
Comments and Suggestions for Authors
Dear authors,
The Eastern Mediterranean is subject to an increase in desert dust pollution over the last several decades. Consequently, there are accordingly responses of surface water temperature and atmospheric humidity to this increased dust pollution condition. Using satellite and in-situ observations, this study investigated the influence of a severe dust event on the diurnal behavior of surface water temperature and absolute humidity in the Kinneret Lake. However, it is only a case study. The analyses are shallow. I could not recommend it for publication. Suggestions for improvement of this manuscript are listed below.
Major comments:
1. The significance of this study is weak; the authors should explain why their investigation is important, and what are the potential application of their conclusions more in detail.
2. How many meteorological station data were used in this study? It was three in Abstract, but only two are introduced in Section 2.3.
3. The authors discussed the heat fluxes from the subsurface to surface are not a causal factor for the air humidity, but the only evidence is the differences of SSWT and SWT. Moreover, the sensible and latent heat fluxes, were not considered.
4. The decrease of 9°C in SWT in Kinneret Lake co-occurred with the dust event, and it does make sense that the dust event could lead to a decrease in SWT, but it can not conclude that the 9°C cooling is contributed by only the dust event without other impacts (e.g., wind speed).
5. As a case study, it is better to give the underlying mechanisms and quantitative estimation of different factors that contribute to the decrease in SWT and increase in air humidity.
Comments on the Quality of English Language
Specific comments:
1. Abstract, Line 13, I suggest use September rather than Sept. throughout the manuscript.
2. Abstract, Line 11, “Using satellite and in-situ observations, …”
3. Introduction, Line 57-59, why it is critical to conduct a comprehensive investigation of the influence of dust events on SWT in Lake Kinneret?
4. Figure 1, it is better to put the station names of Zemah and Bet Dagan in the map.
5. Figure 1, please add more tick labels in Fig.1a.
6. Figure 2, please add the longitude and latitude, and mark the Lake Kinneret in the maps.
7 Section 2.2, Line 92, LT means local time? And please indicate was it am or pm?
8. Section 2.3, Line 142-147, the longitudes and latitudes of the Zemah and Bet Dagan stations, as well as the variables provided and used in this study are not clearly descript.
9. Section 2.3, Line 148-149, were the time series been processed as hourly mean?
10. Section 3.1, Line 152, what means SR? has not been mentioned before.
11. Section 3.1, Line 156, what means global SR? Here global maybe confused and need further explanation.
There are also some other minor mistake or unclear representation, but I think it is more important to address the major comments, after that this study may deserve a reconsideration.
Reviewer 3 Report
Comments and Suggestions for Authors - Overall:
This manuscript analysed the patterns of the water surface temperature of a lake during the passage of a dust storm. Despite the interest in understanding the lake behaviour during such events (especially evaporation), I do not fully agree with the methodology used for this purpose. I did not understand the point of comparing the difference between SSWT measured at a point with the mean value of SWT over a large area, with difference provided being not really plausible. The authors also use water evaporation as a possible reason for this difference, but they do not provide any data regarding this variable.
I invite the authors to further work on their manuscript and resubmit it, following my questions and recommendations below.
- Major issues:
There is no methodology regarding the number of pixels (the criteria) used in the extraction of the hourly SWT data, and considering there is a duststorm, this should be explained. From the images in Fig. 2, there were no pixels in the image from Sep 7 to Sep 9, while in Fig. 6 you showed the complete hourly SWT data. Are you sure no dust pixels were used? In Fig. 9 it shows a difference between SSWT and SWT of up 12ºC, that doesn't really make sense. And how homogeneous was SWT over the polygon, did it allow for comparison with the point measurement of SSWT?
The discussion of water temperature did not include air temperature at all, which is the major driver of water surface temperature.
The inference of evaporation being the driver of temperature fluctuations could be actually tested with calculating the latent heat flux during this short period of time. Presenting the relative humidity could also indicate the possible amount of evaporation occuring during this time period.
The plots used in the manuscript also need to be redone.
Reviewer 4 Report
Comments and Suggestions for Authors
The following is the review for:
Impact of a severe dust event on diurnal behavior of surface water emperature and absolute humidity in subtropical Lake Kinneret.
The paper focuses on how a severe dust event can impact the diurnal variability of surface water temperature. The works focuses on three days in September 6,7,8,9 of 2015. The paper applies Geostationary satellite data to exaimine the specific impacts on the remote sensing product. Data is also examined at multiple meteorological states. The Lake is chosen because it is impacted by the Saharan Dust storms. The paper goes into detail about the changes in both surface water temperature as well as temperature at depth. The authors also examine the impacts on absolute humidity and relate changes in surface water temperature and and absolute humidity to incoming solar radiation as well as evaporation.
Overall the paper is well written and organized. The paper presents an interesting problem with respect to possible errors in satellite derived SST due to dust storms and possible aerosols. I believe the paper should be published after perhaps a minor revision. I really only have one major comment:
I believe the authors need to go into more detail on the impact to satellite derived SSTs. One question would be as infrared derived satellite SSTs apply cloud masking, would this also not filter out the dust storms? My understanding is that there is considerable ongoing research going on in trying to correct satellite derived SSTs for aerosols? How would this impact this study? In Figure 6 how were SSTs derived from METEOSAT under the dust conditions? Assume no aerosol correction was applied. My one recommendation to the authors is to go in more detail about the corrections applied to the METEOSAT data. Assume that for all these September days the application of the cloud masking indicated cloud free conditions.
With respect to the English overall the paper should just need a final proofread.
Comments on the Quality of English Language
English is fine.